# Further Delineation of Duplications of *ARX* Locus Detected in Male Patients with Varying Degrees of Intellectual Disability

**DOI:** 10.3390/ijms23063084

**Published:** 2022-03-13

**Authors:** Loredana Poeta, Michela Malacarne, Agnese Padula, Denise Drongitis, Lucia Verrillo, Maria Brigida Lioi, Andrea M. Chiariello, Simona Bianco, Mario Nicodemi, Maria Piccione, Emanuela Salzano, Domenico Coviello, Maria Giuseppina Miano

**Affiliations:** 1Institute of Genetics and Biophysics “Adriano Buzzati-Traverso”, National Research Council (CNR), 80131 Naples, Italy; poetaloredana@libero.it (L.P.); agne-88@hotmail.it (A.P.); denise.drongitis@igb.cnr.it (D.D.); lucia.verrillo@igb.cnr.it (L.V.); 2Department of Science, University of Basilicata, 85100 Potenza, Italy; maria.lioi@unibas.it; 3Institute for Electromagnetic Sensing of the Environment (IREA), National Research Council (CNR), 80124 Naples, Italy; 4Laboratory of Human Genetics, Scientific Institute for Research, Hospitalization and Healthcare (IRCCS), Giannina Gaslini Institute, 16147 Genoa, Italy; michelamalacarne@gaslini.org (M.M.); domenicocoviello@gaslini.org (D.C.); 5Department of Physics, University of Naples “Federico II”, 80126 Naples, Italy; chiariello@na.infn.it (A.M.C.); simona.bianco@na.infn.it (S.B.); nicodem@na.infn.it (M.N.); 6National Research Council-Superconducting and Other Innovative Materials and Devices Institute (CNR-SPIN), Unit of Naples, 80126 Naples, Italy; 7Department of Health Promotion, Mother and Child Care, Internal Medicine and Medical Specialties, University of Palermo, 90127 Palermo, Italy; maria.piccione@unipa.it; 8Medical Genetics Unit, Azienda Ospedaliera Ospedali Riuniti (AOOR), Villa Sofia Cervello, 90146 Palermo, Italy; salzanoemanuela@gmail.com

**Keywords:** Xp21.3 duplication, *ARX*, intellectual disability, ultraconserved enhancers, KDM5C-SYN1 axis, 3D structure

## Abstract

The X-linked gene encoding aristaless-related homeobox (*ARX*) is a bi-functional transcription factor capable of activating or repressing gene transcription, whose mutations have been found in a wide spectrum of neurodevelopmental disorders (NDDs); these include cortical malformations, paediatric epilepsy, intellectual disability (ID) and autism. In addition to point mutations, duplications of the *ARX* locus have been detected in male patients with ID. These rearrangements include telencephalon ultraconserved enhancers, whose structural alterations can interfere with the control of *ARX* expression in the developing brain. Here, we review the structural features of 15 gain copy-number variants (CNVs) of the *ARX* locus found in patients presenting wide-ranging phenotypic variations including ID, speech delay, hypotonia and psychiatric abnormalities. We also report on a further novel Xp21.3 duplication detected in a male patient with moderate ID and carrying a fully duplicated copy of the *ARX* locus and the ultraconserved enhancers. As consequences of this rearrangement, the patient-derived lymphoblastoid cell line shows abnormal activity of the ARX-KDM5C-SYN1 regulatory axis. Moreover, the three-dimensional (3D) structure of the *Arx* locus, both in mouse embryonic stem cells and cortical neurons, provides new insight for the functional consequences of *ARX* duplications. Finally, by comparing the clinical features of the 16 CNVs affecting the *ARX* locus, we conclude that—depending on the involvement of tissue-specific enhancers—the *ARX* duplications are ID-associated risk CNVs with variable expressivity and penetrance.

## 1. Introduction

The aristaless-related homeobox gene (*ARX*; MIM 300382), which belongs to the aristaless-related subset of the paired (Prd) class of homeodomain proteins, encodes an essential transcription factor (TF) involved in the development of the mammalian cortex [1]. In doing this, ARX controls numerous functions such as neuronal stem-cell proliferation, migration and differentiation, axonal guidance, and synaptic activity [2,3,4,5,6,7]. As *ARX* is located in Xp21.3, loss-of-function (LoF) and partial-LoF mutations contribute to a nearly continuous series of X-chromosome-linked neurodevelopmental disorders (NDDs) ranging from lissencephaly with abnormal genitalia (XLAG; MIM 300215), developmental and epileptic encephalopathy type 1 (DEE1; MIM 308350), X-linked intellectual disability (XLID; 300419), ID with hand dystonia (Partington syndrome; MIM 309510), and autism [8,9,10,11,12]. Of note, many target genes of this pleiotropic TF have been correlated with ID and autism, emphasizing the importance of ARX for brain development and functioning [7,9,13]. Indeed, as previously shown by us, ARX regulates the transcription of the X-linked NDD gene Lysine-specific demethylase 5C (*KDM5C/JARID1C/SMCX*; MIM 314690). We also established that ARX belongs to a convergent transcriptional axis that involves other NDD genes such as Synapsin I (*SYN1;* MIM 313440) and Sodium channel neuronal type II alpha subunit (*SCN2A;* MIM 182390) [9,14,15]. Most importantly, it has been suggested that the tissue-specific activities of ARX are determined by physical interactions with other transcription factors, such as MyoD and Mef2C [16]; with cofactor proteins, such as Groucho/transducin-like enhancer split protein (TLE); or with Wnt signaling [17,18]. In mice, the complete knockdown of *Arx* results in a severe defective corticogenesis with a phenotype similar to the clinical manifestations observed in XLAG individuals with *ARX* LoF mutations [19,20]. Unlike the effects caused by point mutations, it is unclear how duplications of the *ARX* locus can interfere with brain development and functioning. Some studies proposed that *ARX* duplications do not have detrimental effects on the brain. Thus, the abnormal neurological phenotype was ascribed to the presence of an extra copy resulting from the effect of other not-yet-identified chromosomal or molecular anomalies, alone or in association with the *ARX* duplication [21]. However, other studies proposed that the genomic duplications of the *ARX* locus could affect an auto-regulatory feedback loop of *ARX* mediated by ultraconserved long-range enhancers [22,23]. In mice, these regulatory regions are specifically active in the forebrain; furthermore, animals with single or pairwise deletions of these regions showed structural brain defects and alterations of specific neuron populations [24]. Based on these findings, we cannot exclude that copy-number gain may alter the expression level of *ARX* in specific brain areas during embryonic development. Furthermore, as witnessed by several chromosome conformation capture (3C) studies, genomic duplications alter the chromatin portioning units, leading to the formation of new topologically associated domains (neo-TAD) [25,26,27]. Mechanistically, changes in the genomic architecture of a developmentally regulated gene could disturb the activity of its regulatory elements and cause developmental disorders with variable expressivity. Remarkably, duplications of the Xp21.3 region, where *ARX* is located, were found in male patients presenting varying degrees of cognition defects [21,22,28,29,30,31]. Moreover, the correlation of genotype–phenotype was not fully resolved because, out of the 15 Xp21.3 duplications described in the literature, two were detected in two individuals with normal intelligence [21]. The unpredictable and variable phenotypic outcome associated with Xp21.3 duplications poses an enormous diagnostic difficulty to the clinician and, as such, it challenges the general paradigm according to which rearrangements involving dose-sensitive genes may induce a disease. In this framework, finding out whether ARX-target genes and related molecular pathways are sensitive to the amount of ARX would be an important step in understanding the molecular effects caused by an extra copy of this crucial brain TF. Here, we provide an overview of the 15 *ARX* duplications reported in the literature and one new case identified by us. We also propose that ARX is a dosage-sensitive regulator of the target genes involved in neuronal maturation [9,15], providing an entry point to study the molecular effects resulting from *ARX* duplication. We also reconstructed, for the first time, the three-dimensional (3D) structure of the *Arx* locus, both in mouse embryonic stem cells (ESC-46C) and murine cortical neurons (CNs); this allowed us to identify the TAD features underlining changes during neuronal cortical commitment. In the context of ID pathology, understanding the impact of genomic structural variants on the transcriptional regulation of ARX could aid in the interpretation of rare structural variants affecting this locus, and ameliorate the diagnosis and classification of *ARX*-related phenotypes.

## 2. Duplications of the p23.1 Region of the X Chromosome

Segmental duplications of the p21.3 region on the short arm of X chromosome—where *ARX* is located—have been identified in several patients with ID [32]. They are rare, vary in size (from a few kilobases to many megabases) and their clinical evaluation remains debated [32]. To date, 15 duplications of Xp23.1 have been reported (Figure 1 and Table 1). Whibley et al. [32] describe a 41 kb duplication, maternally inherited, identified in a male patient with moderate ID (family 505). This duplication, in direct tandem orientation, includes the terminal exon of Polymerase DNA alpha-1 (*POLA1*; MIM 312040), a ubiquitously expressed gene encoding the catalytic subunit of DNA polymerase, and across the entire coding sequence of *ARX* (Table 1). Of note, mutations in *POLA1* have been found in patients presenting a multisystemic disorder without neurological symptoms, named X-linked reticulate pigmentary disorder (PDR; MIM 301220) [33].

Popovici et al. [21] reports on four duplications of Xp21.3. Two of them, a 440 Kb and a 580 Kb duplication, included the full copies of *POLA1* and *ARX* genes and were found in two males with syndromic ID (case 2 and case 4; Table 1); meanwhile, the remaining two—a 540 Kb duplication, which included a full copy of the *ARX* gene, and a 630 Kb duplication, which included full copies of Phosphate cytidyltransferase 1 choline beta isoform (*PCYT1B*; MIM 300948), *POLA1* and *ARX*—were found in two males with diagnoses of normal intelligence (case 1 and case 3; Table 1). In addition, the authors define two other duplications reported in public databases: a 720 Kb de novo duplication that contained a full copy of *PCYT1B*, *POLA1* and *ARX* genes, identified in a boy with syndromic ID (case 5, reported by DECIPHER https://www.deciphergenomics.org/, accessed on 31 January 2022); and a 290 Kb duplication covering partial *POLA1* and *ARX* genes, identified in a patient with developmental delay (reported by the ISCA consortium; case 6; Table 1).

**Table 1 ijms-23-03084-t001:** *ARX* duplications and clinical spectrum.

Patient	Extent (kb)	ChrX Genomic Coordinates (UCSC Release)	Duplicated Genes	Duplicated Enhancers	Clinical Signs/Reference
505	41.1	24,992,915–25,033,979 (GRCh37/hg19) 24,974,798–25,015,862 (GRCh38/hg38)	*ARX* (full gene)	hs121, hs122, hs145	Moderate Syndromic ID [32]
Case 1	540	24,861,402–25,398,496 (GRCh37/hg19) 24,843,285–25,380,379 (GRCh38/hg38)	*ARX* (full gene)	hs118, hs119, hs121, hs122, hs145	No clinical signs [21]
Case 2	440.5	24,593,306–25,033,770 (GRCh37/hg19) 24,575,189–25,015,653 (GRCh38/hg38)	*POLA1* (full gene) *ARX* (exons 2-5)	hs118, hs119, hs121, hs122, hs145	Severe ID, Microphthalmia, growth retardation [21]
Case 3	630	24,537,027–25,163,704 (GRCh37/hg19) 24,518,910–25,145,587 (GRCh38/hg38)	*PCYT1B* (full gene); *POLA1* (full gene); *ARX* (full gene),	hs118, hs119, hs121, hs122, hs145	No clinical signs [21]
Case 4	580	24,650,157–25,230,368 (GRCh37/hg19) 24,632,040–25,212,251 (GRCh38/hg38)	*POLA1* (full gene), *ARX* (full gene)	hs118, hs119, hs121, hs122, hs145	autism, hyperactivity, delayed speech [21]
Case 5	720	24,542,008–25,542,728 (GRCh37/hg19) 24,523,891–25,524,611 (GRCh38/hg38)	*PCYT1B* (full gene); *POLA1* (full gene); *ARX* (full gene),	hs118, hs119, hs121, hs122, hs145, hs123	ID, psychiatric abnormalities, delayed speech [21]
Case 6	290	24,733,304–25,022,540 (GRCh37/hg19) 24,715,187–25,004,423 (GRCh38/hg38)	*POLA1* (partial), *ARX* (partial)	hs118, hs119, hs121, hs122, hs145	Developmental delay [21]
A150	3350	24,513,979–27,864,451 (GRCh37/hg19) 24,495,862–27,846,334 (GRCh38/hg38)	*PCYT1B* (full gene), *POLA1* (full gene); *ARX* (full gene) & others	hs118, hs119, hs121, hs122, hs145, hs123	Autism [30]
DECIPHER 277835	302	24,843,484–25,145,646 (GRCh38/hg38)	*ARX* (full gene)	hs118, hs119, hs121, hs122, hs145, hs123	Moderate ID [30]
DECIPHER 268043	2300	24,810,754–27,125,219 (GRCh37/hg19) 24,792,637–27,107,102 (GRCh38/hg38)	*ARX* (full gene) &others	hs118, hs119, hs121, hs122, hs145, hs123	ID; short stature [30]
DECIPHER 250183	717	24,807,990–25,524,611 (GRCh37/hg19) 24,789,873–25,506,494 (GRCh38/hg38)	*ARX* (full coding region)	hs118, hs119, hs121, hs122, hs123, hs145	Behavioral abnormality; ID; delayed speech [30]
DECIPHER 265145	580	24,632,040–25,212,251 (GRCh37/hg19) 24,613,923–25,194,134 (GRCh38/hg38)	*POLA1* (full coding region) *ARX* (full coding region)	hs118, hs119, hs121, hs122, hs145	attention deficit; hyperactivity; autism; delayed speech [30]
P1	438	24,887,676–25,325,777 (GRCh37/hg19) 24,869,559–25,307,660 (GRCh38/hg38)	*ARX* (full coding region)	hs118, hs119, hs121, hs122, hs145	mild ID, speech delay and hypotonia [22]
P3	377	24,677,441–25,054,698 (GRCh37/hg19) 24,659,324–25,036,581 (GRCh38/hg38)	*POLA1* (full coding region) *ARX* (full coding region)	hs118, hs119, hs121, hs122, hs145	Developmental delay, growth retardation, delayed speech [22]
DECIPHER 266096 DP3	813	24,741,372–25,554,818 (GRCh37/hg19) 24,723,255–25,536,701 (GRCh38/hg38)	*ARX* (full coding region)	hs118, hs119, hs121, hs122, hs145, hs123	hypotonia, microcephaly [22]
GE#1	803	24,828,871–25,631,863 (GRCh38/hg38)	*ARX* (full coding region)	hs118, hs119, hs121, hs122, hs145, hs123	Moderate ID; psychomotor retardation, behavioural and developmental disorder [this report]

Egger et al. [30] describe a de novo 3.3 Mb duplication containing the full copy of *ARX* and other genes, detected in a male patient with autism (A150; Table 1). They also report on four gain CNVs that are present in the DECIPHER database: a 302 Kb duplication (DECIPHER 277835) including the full copy of the *ARX* locus, detected in a male patient with moderate ID; a 2.3 Mb duplication (DECIPHER 268043), identified in a male patient with ID; a 717 Kb duplication (DECIPHER 250183), detected in a male patient with ID and behavioral abnormality; and a 580 Kb duplication (DECIPHER 265145), found in a male patient with autism and delayed speech and language development (Table 1). In Ishibashi et al. [22], three gain CNVs including the *ARX* locus were described: a 438 Kb duplication (P1) including the full copy of *ARX*, detected in a male patient with mild ID, speech delay and hypotonia; a 377 Kb duplication (P3) including a full copy of *POLA1* and *ARX*, found in a male patient with developmental delay, growth retardation, and delayed speech; and a 813 Kb duplication (DECIPHER 266096 DP3), detected in a male patient with hypotonia and microcephaly (Table 1). Additionally, we identified, in a male patient with moderate ID and psychomotor retardation (GE#1; see Appendix A), a new interstitial duplication of Xp22.11-p21.3 with an estimated size of at least 803 Kb including the full copy of *ARX* (Table 1). Summarizing what is described above, except for two cases (case 1 and case 3), all the published duplications and the new one reported here were identified in patients with various neurodevelopmental problems, intellectual disabilities and autism (Table 1) [22,30,32]. Furthermore, all these duplications show different breakpoint positions around the *ARX* locus, and in several cases, they include a gene desert area (1.1 Mb length) located between *ARX* (hg38 chrX: 25,003,694–25,015,965) and the MAGE family member B18 (MAGEB18; hg38 chrX:26,138,343–26,140,736).

## 3. Identification of a Novel Xp22.11-p21.3—Duplication in a Male Child with Moderate ID and Analysis of Functional Implications

A new interstitial duplication of Xp22.11-p21.3 was detected by CGH array in a male child with moderate ID and his asymptomatic mother (family GE#1; Figure 2A). As determined by the UCSC Genome database (GRCh38/hg38 release), this rearrangement has an estimate size at least of 803 Kb, from nucleotide position 24.828.871 (Xp22.11) to 25.631.863 (Xp21.3) (Figure 2A,B). By inspecting the sequence content, we established that this interstitial duplication contained a partially duplicated *POLA1* locus, an additional copy of the *ARX* locus (including intact duplicated promoter sequence and all enhancer elements) and a non-actively transcribed gene region (Figure 2B). We analyzed the effect of this rearrangement in the lymphoblastoid cell line derived from the ID patient (ID-LCLs) compared to LCLs derived from the control XY individual. Accordingly, with the presence of two active gene copies of *ARX*, we observed about a 3,5-fold increased expression of the *ARX* transcript with respect to control cells (Figure 2C). On the contrary, no difference in the level of *POLA1* expression was found between the control and patient cells (Appendix A), indicating that this gene is not involved in the disease phenotype.

It has been estimated that gene dosage alterations caused by structural variations are responsible for ~15% of NDD cases [34]. Given the extra dosage of *ARX* transcript, we evaluated the expression levels of two ARX-related disease genes involved in ID and epilepsy: *KDM5C*, a direct target of ARX—as already established by us in both human and murine cell systems [9,14,35]—and *SYN1*, a gene found physiologically repressed by KDM5C [14,15]. A marked downregulation of *KDM5C*/KDM5C, both at mRNA and protein levels (Figure 2C and Appendix A), coupled with the overexpression of *SYN1*, was observed in ID-LCLs with respect to the control (Figure 2C). As ARX can activate or repress target promoters [7,9], we next analyzed whether the transcriptional activity of ARX operates in a dosage-dependent manner. By means of a luciferase assay performed in HeLa cells, we tested the promoter response of *KDM5C*, varying the amount of the transfected construct expressing the *ARX* WT cDNA (*ARX*_Myc-WT). Thus, an inverse correlation between the reporter response of the *KDM5C* promoter (JD_full_Luc) and the ARX dosage was found (Figure 2D). This evidence suggests that ARX may act on the *KDM5C* promoter as repressor at a high concentration, and as activator at a low concentration (Figure 2D). Next, we verified whether this double function of ARX on *KDM5C* is cell-context dependent by testing the transient activity of ARX on the *KDM5C* promoter in two different human cell lines. We found that a low concentration of *ARX* [25 ng] transiently transfected in HEK293T and SH-SY5Y induces the repression or the stimulation of *KDM5C* promoter, respectively (Figure 2E), indicating that the bi-functional activity of ARX is dose- and cell-context dependent.

## 4. Ultraconserved *ARX/Arx* Brain Enhancers

Similarly to other developmentally expressed TF loci, the *ARX* locus has a high density of multiple and very long human/rodent ultraconserved enhancers [23,24,36,37,38,39]. As annotated in the VISTA enhancer database, all these regulatory elements, except one, are located downstream of the coding sequence of *ARX*, between the *ARX* and *POLA1* loci (Figure 3A) [36]. In the mouse forebrain, four of them (hs119, hs121, hs122 and hs123) showed enhancer activity similar to the *Arx* gene expression (Figure 3B). Of note, mice with single or pairwise deletions of ultraconserved *Arx* enhancers presented neurological or growth abnormalities (Figure 3C) [24]. In particular, the loss of enhancer hs121 (hs121-KO) caused altered densities of specific cortical interneuron types, including cholinergic neurons; meanwhile the loss of enhancer hs122 (hs122-KO), which is active in the dorsal forebrain, caused a reduced expression of *Arx* in the dorsal forebrain and changes in the size and morphology of the hippocampus. Regarding hs119, its deletion caused growth abnormalities and reduced body mass (Figure 3C) [24]. Very interestingly, hemizygous double-knockout mice for hs119 and hs121 (hs119/hs121-KO) displayed a more severe combination of defects when each enhancer was deleted individually (Figure 3C). Furthermore, several of the neuronal alterations observed in single- and double-knockout mice were very similar to a subset of defects observed in *Arx* KO mice [3,19,38,39]. These findings sustain not only the crucial role of these enhancers in the regulation of *Arx*, but also reinforce the concept that rearrangements involving the *ARX* locus could interfere with their regulatory activity. Moreover, a functional analysis carried out in zebrafish revealed that three of the ultraconserved enhancers mediate negative and positive autoregulation of *ARX* in specific brain regions [22]. The authors proposed a model, explaining how breakpoints in long-range enhancers might alter the expression levels of *Arx* in specific brain regions, and how this effect can lead to subtle neuronal phenotypes [22]. With this in mind, we therefore conclude that perturbations of long-range activity of *ARX/Arx* ultraconserved enhancers could have dosage-sensitivity-driven effects, and thus, constitute a risk for neurodevelopmental disorders.

## 5. Three-Dimensional (3D) *Structure* of the *Arx* Locus in Embryonic Stem Cells (ESs) and Cortical Neurons (CNs)

The alteration of the 3D structure of chromosomes is emerging as one of the causes of ID [25]. Mechanistically, when chromosomal rearrangement events affect TAD boundaries, they alter the interactions between genes and enhancers, leading to an abnormal expression of genes. Given the complexity of the *ARX/Arx* locus, we investigate its architecture, analyzing published murine HiC data in embryonic stem cells (ESs) and cortical neurons (CNs) [40]. Thus, we focused on a 5 Mb genomic region (chrX:91,000,000–96,000,000, UCSC mm10) encompassing the *Arx* gene and its ultraconserved enhancers [23]. From the analysis of the contact maps, it emerges that the locus undergoes structural rearrangements involving the *Arx* gene and the enhancer elements, highlighting how architectural features have a strong functional impact on the regulation of this gene (Figure 4). In ESs, the *Arx* gene tends to interact locally with its most proximal enhancers (hs119, hs121 and hs122) and interacts weakly with its flanking regions (Figure 4B, left matrix). In CNs, in addition to the contact with the most proximal enhancers (hs119, hs121 and hs122), we observed long-range contacts with the upstream region involving the most distant hs123 ultraconserved enhancer, located in a large, well-defined interaction domain (Figure 4B, right matrix). These features are compatible with a transcriptional activity change in *Arx*; it is inactive in ESs, and active in CNs in which all four of the ultraconserved enhancers cooperate to control the *Arx* expression [24]. Next, we investigated the 3D structure of the locus in both cell lines by employing the Strings and Binder Switch polymer model [41,42,43,44]. The model assumes that the chromatin filament (the string) has a specific binding site that interacts with molecular factors (the binders), which are naturally present in the nuclear environment. Moreover, the interaction between binding sites and binders forms stable loops, and shapes the locus organization (see Appendix A). The contact maps obtained from the model effectively recapitulate the interaction pattern of experimental data (Pearson r = 0.98, Figure 4A, bottom matrices). Regarding the distribution of the binding sites, the studied region (Figure 4B) reveals specific architectural features: in ESs, the *Arx* gene is associated to a binding domain (number 9) containing the hs119, hs121 and hs122 enhancers, but not the hs123; in CNs, this binding domain changes and spreads, involving the hs123 enhancer, in agreement with HiC data. In order to give a structural rationale to these findings, we performed Molecular Dynamics (MD) simulations and reproduced the 3D architecture in both ES and CN cells (Figure 4C). For visualization purposes, we divided the polymer in three consecutive regions; then, we colored them according to the binding domain distributions. We found that in ESs the three regions tend to form locally self-interacting domains, weakly interacting with each other. Conversely, in CNs, we found that the green and blue regions interact strongly in order to bring the enhancer hs123 into close proximity with *Arx*, ensuring its activity (Figure 4C). Based on this evidence, we therefore proposed that the long-range interactions between the regulatory elements of *Arx* and its promoter could differently regulate the expression of this pleiotropic TF during embryonic stem-cell proliferation and neuron differentiation. Very interestingly, comparison of *ARX* expression data across brain development in mice, humans and human-derived brain organoids, available in the Human and Mouse Development atlas (http://www.humous.org/, accessed on 2 February 2022) [45], revealed a distinctive time- and cell-dependent expression pattern in the mouse neocortex, human neocortex and human-derived brain organoids (Figure 5A–C). In particular, in the murine cortex, *Arx* is strongly expressed in apical radial glia cells (aRGs) at E12, and is decreased in immature neurons (iNs) and mature neurons (mNs), along the following corticogenesis stages (E15; Figure 5A). Conversely in the human cortex, *ARX* is strongly expressed in basal radial glia cells (bRGs) and iNs at gestation week 8, and decreased along the following embryonic stages (at week 20; Figure 5B); meanwhile in human-derived brain organoids *ARX* is expressed in RG cells at the early phase of in vitro culture (at week 3; Figure 5C). Based on these findings, we therefore conclude that the fine regulation of *ARX* expression in time and space mirrors the architectural features of its genetic locus and the relative regulatory elements.

## 6. Genotype-Phenotype Correlation

Sixteen distinct Xp21.3 duplications with different breakpoints, ranging in size from 41.1 kb to 3.3 Mb, have been identified to date (Table 1). Clinical findings in Xp21.3 duplication include syndromic ID (from mild–moderate to severe), global developmental delay, hypotonia, autism, and hyperactivity (Table 1). The majority of these rearrangements include a full copy of *ARX* with the telencephalon enhancers, and a full copy or break of further genes. Remarkably, the smallest one—identified in a patient with moderate syndromic ID (Patient 505) [32]—includes the full copy of *ARX* and the enhancer elements hs121, hs122 and hs145 (Table 1). In contrast, large duplications include full copies or breakages of *PCYT1B* and/or *POLA1*, two genes not associated with central nervous system disorders [33]. The molecular characterization of these rearrangements suggests that the core of ID syndrome is largely due to the duplication of the *ARX* locus. In this context, the activity of the duplicated telencephalon enhancers could contribute to the neurological features. Indeed, similarly to other developmentally expressed genes with pleiotropic activity, *ARX* has multiple enhancers orchestrating its regulation in time and space in the developing brain [23,24,36,37,38,39]. Thus, a structural alteration within these regions can specifically disturb the transcriptional regulation of *ARX*. Starting with these considerations, we propose that the ID variability observed in patients with Xp21.3 duplication could be due to disturbed cis-regulatory mechanisms of *ARX* expression that, in turn, lead to different syndromic conditions. Particular consideration should be given to the two duplications reported as asymptomatic by Popovici et al. [21] (case 1 and case 3), which raised the general question of whether the duplications of Xp21.3 are pathogenetic. As already described, two male individuals with normal IQ, carrying two different duplications of 540 kb (case 1) and 630 kb (case 3), were identified 21]. Both rearrangements enclosed the *ARX* gene and the enhancer elements hs118, hs119, hs121, hs122 and hs145 [21]. Since this region has been found duplicated in several male patients with varying degrees of ID and autism, it does not seem plausible that these rearrangements are completely asymptomatic. As previously commented by others, the presence of these two duplications could be attributable to an untested somatic mosaicism condition [22]. Consequently, the two rearrangements could be absent in the brain tissues or present in a small fraction of cells, and thus, could not disturb the control of *ARX* expression in the developing brain. As an alternative explanation, we cannot rule out that the non-ID phenotypes associated with these two duplications could be the result of variable phenotypic penetrance, a feature often occurring in syndromic ID conditions. Altogether, these considerations strongly support the concept that Xp21.3 duplications are ID-associated risk CNVs with variable expressivity and incomplete penetrance. Certainly, all patients with Xp21.3 duplication should receive a thorough genetic and neurological evaluation in order to avoid neglecting early treatments that could help manage the disease. In this regard, it should be reiterated that the close alliance between clinicians, scientists and families could improve both the characterization of the ID phenotypes and the management of patients.

## 7. Discussion & Conclusions

Patients with various neurodevelopmental problems, intellectual disabilities and autism, were identified to have duplications encompassing the *ARX* gene locus. Although *ARX* rearrangements are rare, thus making access to a wide number of cases difficult, the identification of new cases with CNVs in Xp21.3 allows us to improve our knowledge on ARX and to stimulate the pursuit of further functional studies. Indeed, in spite of the progress made over the years on the functional impacts of point mutations in *ARX*, the need remains to clarify how duplications of specific *ARX* enhancers can alter its bi-functional dose-dependent transcriptional activity. First evidence suggests that these rearrangements may disturb the pleiotropic tunable bifunctional activity of ARX, and thus, may cause neurodevelopmental defects. Indeed, these gain CNVs seem to affect both the protein-coding portion of *ARX* and the copy number and/or the position of the telencephalon enhancers. As a result, they can reshuffle the higher-order chromatin structure of the *ARX* locus. Remarkably, although enhancer duplications may impact gene expression in unpredictable ways [46], structural variants can disrupt the 3D structure of a gene locus, modifying the spacing between enhancers and their position relative to the promoter [47]. Since the level of expression of *ARX* is highly sensitive to the genomic architecture, as observed in mice [24] and GE#1 patient LCLs, we hypothesize that changes in the genomic architecture of the *ARX* locus may dramatically impact its expression levels and, in turn, on the activity of its protein product. Moreover, the 3D structure of the *Arx* locus revealed differences between mouse ESs and cortical neurons in the long-range interactions of the enhancer elements with its promoter. Given the multiple activities of ARX in the brain [48], perturbations of its transcriptional regulation during the critical phases of embryonic development—such as RG proliferation, and neuron differentiation and maturation—could alter the cortical multilayers [1,2,3,4,5,6,7].

In conclusion, duplications of the Xp21.3 region can lead to cognition defects by disrupting the 3D structure of the *ARX* locus and altering the dose-sensitivity action of *ARX*, which is crucial for corticogenesis [1,2,48]. Very importantly, further studies aimed at integrating clinical data with genetic and molecular findings are required to dissect the specific functional processes that are damaged in patients with Xp21.3 duplication, providing prognostic indicators and therapeutic perspectives.

## Figures and Tables

**Figure 1 ijms-23-03084-f001:**
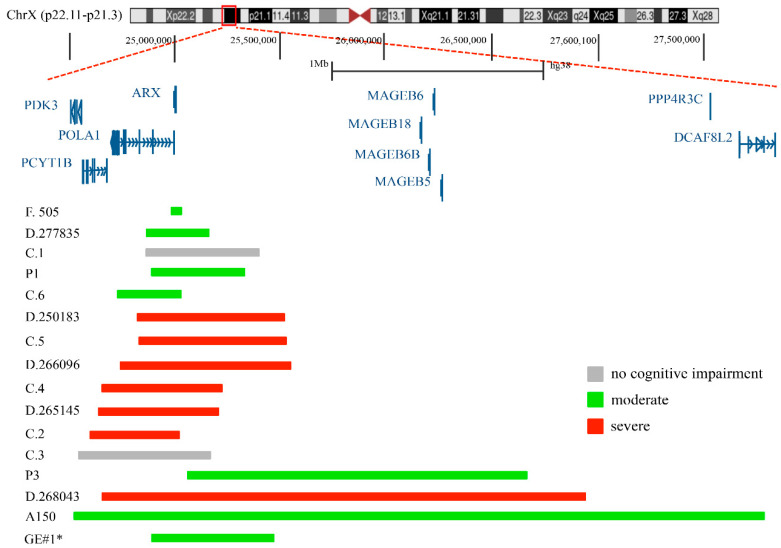
Genomic mapping of duplications encompassing the *ARX* locus. Of them, 15 have already been described in the literature, and one, GE#1*, is described in this report for the first time.

**Figure 2 ijms-23-03084-f002:**
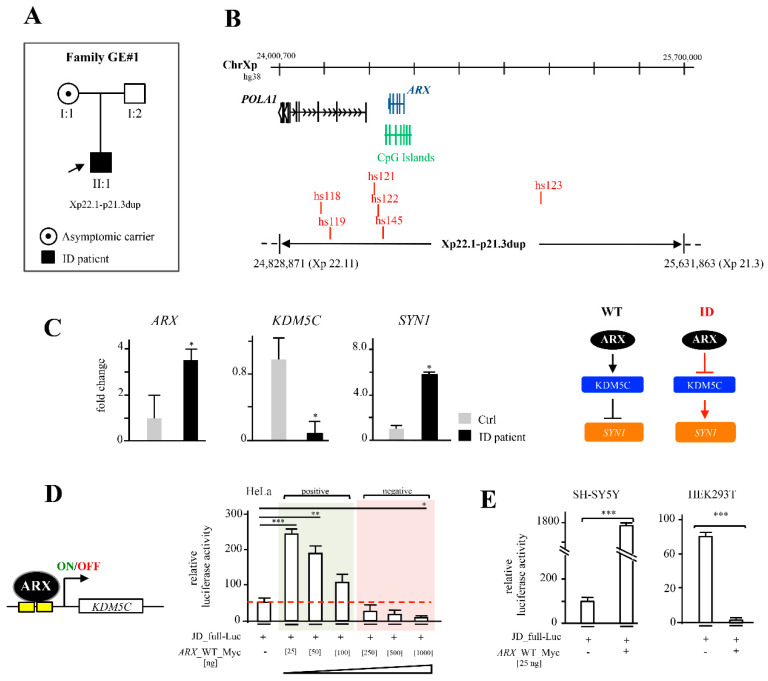
Identification of a novel *ARX* duplication and its functional implications. Genealogical tree of family GE#1 and segregation of Xp22.1-p21.3dup: (**A**) Physical map of the *ARX* duplication on the human X chromosome (ChrX). (**B**) The involved genes *POLA1* and *ARX*, the CpG Islands and the ultraconserved enhancers expressed in embryonic mouse brain (VISTA Enhancer Browser at https://enhancer.lbl.gov/, accessed on 24 January 2022) are shown. *ARX, KDM5C* and *SYN1* expression levels in GE#1 ID patient’s lymphoblastoid cell lines (LCLs) and XYWT LCLs (**C**). Results were normalized using *HPRT1* as reference gene and shown as mean ± SD from three independent experiments. *p* < 0.05. Schematic representation of ARX-KDM5C-SYN1 regulatory path is shown. Dose- and cell-type-dependent activity of ARX on 5′ *KDM5C* regulatory region (**D**,**E**). Luciferase assay in HeLa cells at increasing concentrations of *ARX* and Luciferase assays in HEK293T and SHSY5Y cell lines. Data were normalized to relative luciferase activity of JD-full-Luc. Results are shown as mean ± SD from three independent experiments. *p* < 0.05 *; *p* < 0.005 **; *p* < 0.0005 ***.

**Figure 3 ijms-23-03084-f003:**
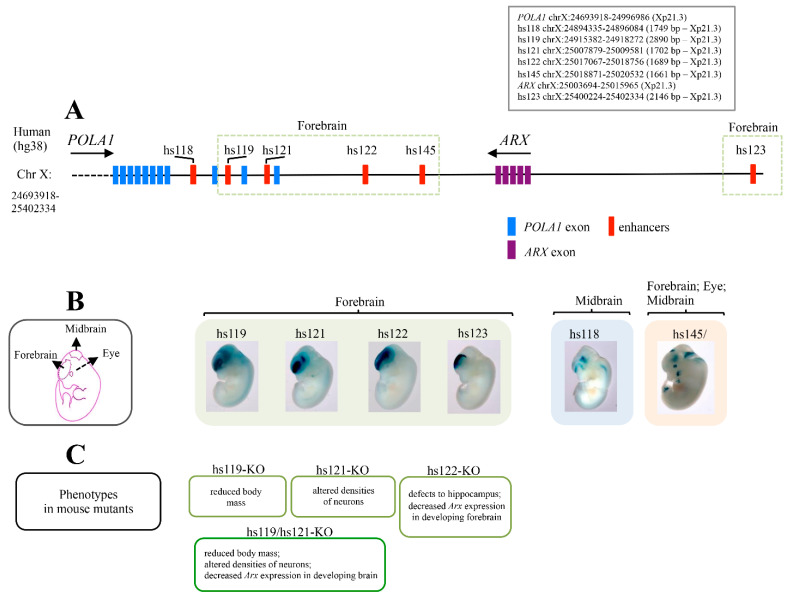
The *ARX* locus has a high density of forebrain ultraconserved sequences. Schematic representation of Xp region containing the multiple ultraconserved sequences located near the *ARX* locus, created on the annotation of UCSC hg38 assembly. Nucleotide positions are shown in box (**A**). Whole-mount staining images, obtained freely from VISTA Enhancer Browser (https://enhancer.lbl.gov/, accessed on 24 January 2022), showing the expression profiles of each enhancer in E11.5 embryos. Schematic representation of E11.5 brain subregions is shown in (**B**). Summary of phenotypic and molecular features of knockout mice lacking individual, or combinations of, ultraconserved *Arx* enhancers in shown in (**C**).

**Figure 4 ijms-23-03084-f004:**
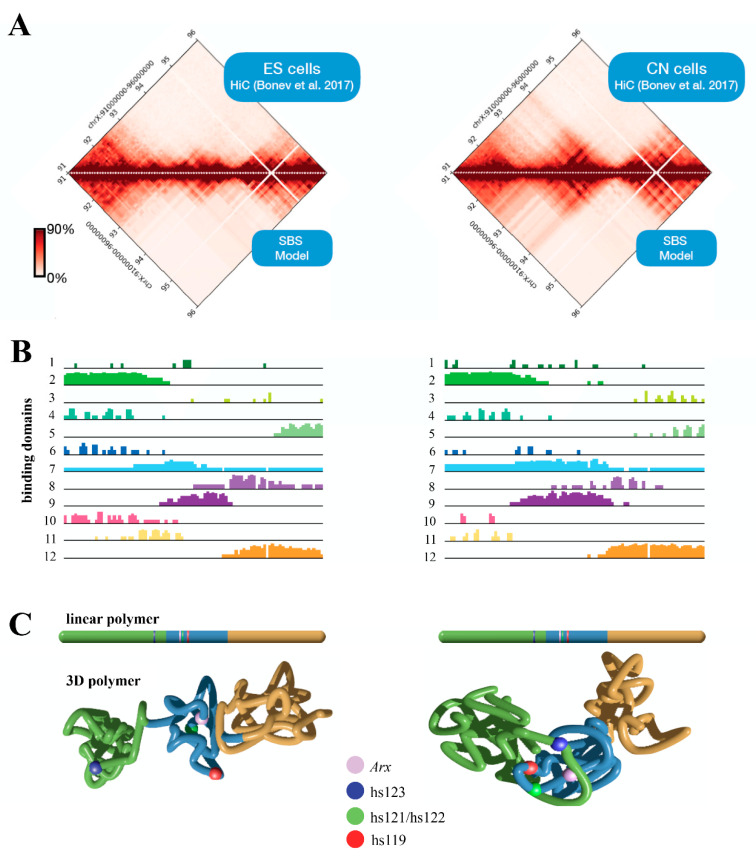
Polymer physics description of the murine *Arx* locus. Bottom matrices and contact maps predicted from the SBS polymer model by analysing the genomic region chrX:91,000,000–95,000,000 (UCSC, mm10). HiC data were obtained from a previous study [20] (**A**). Binding domain distributions for ESs (left) and CNs (right) (**B**). Snapshots from real MD simulations describing the 3D structure in ESs (left) and CNs (right). The colour scheme used is reported in the linear bar above (**C**).

**Figure 5 ijms-23-03084-f005:**
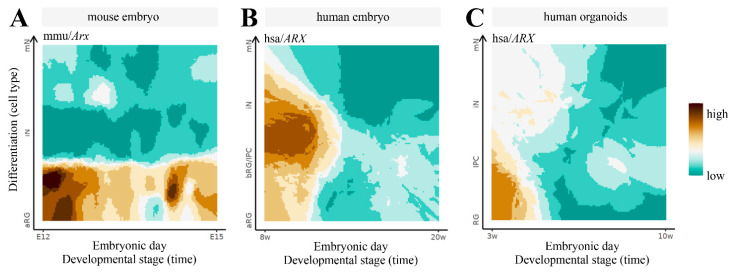
Spatiotemporal gene expression trajectories of *Arx/ARX* across the corticogenesis: in mouse, from the embryonic stage E12 to E15 (**A**); in humans, from gestation week 8 to 20 (**B**); and in human-derived brain organoids, from in vitro developmental week 3 to 10 (**C**). RG, radial glia; aRG, apical radial glia; vRG, ventral radial glia; oRG, outer radial glia; IPC, intermediate progenitor cell; BP, basal progenitor; N, neuron; iN, immature neuron; mN, mature neuron. Data can be accessed freely on http://www.humous.org/, accessed on 2 February 2022.

## Data Availability

Publicly archived datasets analyzed: DECIPHER database https://www.deciphergenomics.org/, accessed on 31 January 2022; Human and Mouse Development atlas http://www.humous.org/, accessed on 2 February 2022.

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
