# Peer review of "Further Delineation of Duplications of ARX Locus Detected in Male Patients with Varying Degrees of Intellectual Disability"

_ijms, 2022, doi:10.3390/ijms23063084_

Round 1
Reviewer 1 Report
The manuscript “Further delineation of duplications of ARX locus detected in male patients with varying degree of intellectual disability” is clearly and well prepared meta-analysis. However, the authors must provide some minor revision:
- Correct from 3 further all the numbers of headings because are the same in line 168 ( Identification of a novel …) and line 220 (3. Ultraconserved ARX/Arx …)
- Please indicate the sources from the Figure 3
Author Response
Reviewer 1 Comments
The manuscript “Further delineation of duplications of ARX locus detected in male patients with varying degree of intellectual disability” is clearly and well prepared meta-analysis.
Response: Thank to the Reviewer1 for this comment.
Minor revision:
Point1. Correct from 3 further all the numbers of headings because are the same in line 168 (Identification of a novel …) and line 220 (3. Ultraconserved ARX/Arx …)
Response1. We thank the referee for noticing these errors. We have them in the MS_R1.
Point2. Please indicate the sources from the Figure 3.
Response2. About the Figure 3, we here specified the source of each item:
In A, we developed the scheme of the genomic region according to the position of each element as reported in UCSC hg38 assembly.
In B, the whole-mount staining images were obtained from the public dataset VISTA Enhancer Browser showing the expression profiles of each enhancer in E11.5 embryos. The data are freely accessed on https://enhancer.lbl.gov/, as specified in the legend of the figure;
In C, we summarized in a schematic representation the phenotypic and molecular findings of the knockout mice lacking individual or combinations of ultra-conserved Arx enhancers, as described in Bejerano et al. 2008, Dickel et al. 2018.
In order to better specify all these and avoid misleading, we changed the legend of Figure 3:
from “Figure 3. The ARX locus has a high density of forebrain ultraconserved sequences. Regions of Xp chromosome containing multiple ultraconserved sequences located near the ARX locus. Nucleotide positions from UCSC hg38 assembly are shown (A). Whole-mount staining images obtained from VISTA Enhancer Browser showing the expression profiles of each enhancer in E11.5 embryos (B). Data can be accessed freely on https://enhancer.lbl.gov/. Schematic representation of E11.5 mouse and subregions were shown. Summary of phenotypic and molecular features of knockout mice lacking individual or combinations of ultra-conserved Arx enhancers (C)” to:
“Figure 3. The ARX locus has a high density of forebrain ultraconserved sequences. Schematic representation of Xp region containing the multiple ultraconserved sequences located near the ARX locus created on the annotation of UCSC hg38 assembly. Nucleotide positions are shown in the box (A). Whole-mount staining images obtained freely obtained from VISTA Enhancer Browser (https://enhancer.lbl.gov/) showing the expression profiles of each enhancer in E11.5 embryos. Schematic representation of E11.5 brain subregions are shown (B). Summary of phenotypic and molecular features of knockout mice lacking individual or combinations of ultra-conserved Arx enhancers (C).
1.0.0.20 1.0.0.20Reviewer 2 Report
This review article by Poeta et al describes the potential correlation between duplication of ARS locus with phenotypic behavior. In this article, author discussed 15 reported cases of ARS duplication, including one identified by them. They identified the new one in a male child with moderate intellectual disability.
Alteration of Ars gene has been associated with neurodevelopmental disorder however, duplication of Ars gene is not completely understood because there are cases where duplication has no effect. This article revolves around exploring genotype-phenotype correlation and postulates different hypotheses like ARS is dose sensitive during neuronal maturation. Overall I enjoy reading this article and it is publishable.
Minor comment
Page 5 Line 168 and page 7 line 220, both heading has same number 3.
Author Response
Reviewer 2 Comments
This review article by Poeta et al. describes the potential correlation between duplication of ARX locus with phenotypic behavior. In this article, author discussed 15 reported cases of ARX duplication, including one identified by them. They identified the new one in a male child with moderate intellectual disability. Alteration of Arx gene has been associated with neurodevelopmental disorder however, duplication of Arx gene is not completely understood because there are cases where duplication has no effect. This article revolves around exploring genotype-phenotype correlation and postulates different hypotheses like ARX is dose sensitive during neuronal maturation. Overall I enjoy reading this article and it is publishable.
Response: Thank to the Reviewer2 for this comment.
Minor comment:
Point1. Page 5 Line 168 and page 7 line 220, both heading has same number 3.
Response. We thank the referee for noticing these errors. We have corrected them in the MS_R1.
1.0.0.20